# Aquafaba from Korean Soybean II: Physicochemical Properties and Composition Characterized by NMR Analysis

**DOI:** 10.3390/foods10112589

**Published:** 2021-10-26

**Authors:** Yue He, Youn Young Shim, Jianheng Shen, Ji Hye Kim, Jae Youl Cho, Wan Soo Hong, Venkatesh Meda, Martin J. T. Reaney

**Affiliations:** 1Department of Chemical and Biological Engineering, University of Saskatchewan, Saskatoon, SK S7N 5A9, Canada; yuh885@mail.usask.ca (Y.H.); venkatesh.meda@usask.ca (V.M.); 2Department of Plant Sciences, University of Saskatchewan, Saskatoon, SK S7N 5A8, Canada; younyoung.shim@usask.ca (Y.Y.S.); jis956@mail.usask.ca (J.S.); 3Prairie Tide Diversified Inc., Saskatoon, SK S7J 0R1, Canada; 4Department of Integrative Biotechnology, Biomedical Institute for Convergence at SKKU (BICS), Sungkyunkwan University, Suwon 16419, Korea; kjhkjhmlml@skku.edu (J.H.K.); jaecho67@gmail.com (J.Y.C.); 5Department of Foodservice Management and Nutrition, Sangmyung University, Seoul 51767, Korea; wshong@smu.ac.kr; 6Guangdong Saskatchewan Oilseed Joint Laboratory, Department of Food Science and Engineering, Jinan University, Guangzhou 510632, China

**Keywords:** aquafaba, soybean, chickpea, physicochemical property, hydration kinetics, NMR

## Abstract

Aquafaba (AQ) emulsification properties are determined by genetics and seed processing conditions. The physicochemical properties and hydration rates of chickpea (CDC Leader) as a control with proven emulsifying properties were recently reported. Here, we identify correlations between soybean (Backtae, Seoritae, and Jwinunikong) physical, chemical, and hydration properties as well as AQ yield from seed and functional (emulsion and foaming) properties. In addition, a total of 20 compounds were identified by NMR including alcohols (isopropanol, ethanol, methanol), organic acids (lactic acid, acetic acid, succinic acid, citric acid, and malic acid), sugars (glucose, galactose, arabinose, sucrose, raffinose, stachyose), essential nutrients (choline, phosphocholine), amino acids (alanine, glutamine), and polyphenols (resveratrol, glycitin). The process used in this study utilizes a soaking step to hydrate the seed of the selected Korean soybean cultivars. The product, AQ, is an oil emulsifier and foaming agent, which is suitable for use as an egg substitute with improved emulsion/foam formation properties when compared with a chickpea-based AQ.

## 1. Introduction

Today, informed consumers demand healthy and nutritious foods. Food oil emulsions, for example, those found in mayonnaise and salad dressing, are often described as potentially unhealthy foods as they are rich in fat and cholesterol. Plant-based proteins, including soybean and white lupin proteins, are potentially useful as replacements for egg-based ingredients in mayonnaise emulsion systems [1,2]. Soybean products, which are used as ingredients in vegan meat substitutes, can contain both high-quality protein and physiologically active substances, such as isoflavones, saponins, phytic acid, phytosterols, and peptides [3]. Isoflavones are a functional component of soybean that can act like estrogen in mammals and, as such, are often classified as phytoestrogens. For example, osteoporosis can be a symptom of lowered estrogen in postmenopausal women and consumption of soy isoflavones can mitigate the effects of lower estrogen levels [4]. In addition, another component of soy, tocopherol, has antioxidant activity [5] and soy peptides impart anti-inflammatory effects [6]. Despite the benefits of consuming soy products, the major soybean products in Asian diets are soymilk, tofu, soy sauce, and soybean paste made from fermented soybean. The development, production, and distribution of new products that extend the use of soy ingredients could lead to greater consumption of this healthy ingredient.

Like other pulses, soybean is usually steeped in water to hydrate the seed, then cooked by either boiling seed in water or cooking it in a pressure cooker to soften the seed and improve digestibility. This process generates large volumes of liquid waste that require treatment before being discarded [7]. The viscous liquid waste separated from canned or pressured-cooked chickpea or other legumes is called aquafaba (AQ) and is a potential eco-friendly nutrient rich by-product [8]. As a food ingredient, AQ imparts excellent foaming, emulsification, and gelling properties to foods. It is often described as an egg replacement in a wide range of new food products. Not only is AQ an inexpensive and accessible resource that can be used as a cholesterol-free egg replacement, but it also provides significant utility to the vegan community because it increases food options for diets free of animal products. Moreover, people who have egg allergies might also benefit from AQ, as it affords more food product options, and at the same time avoids exposure to egg products. Therefore, AQ can be expected to expand the selection of vegetarian-based ingredients with high utility.

However, most research related to AQ components, functionalities, and AQ-based food formulations is limited to AQ produced from chickpea [9,10,11,12,13,14,15,16]. To our knowledge, Serventi et al. (2018) [17] offers the only investigation of the composition and functionality of yellow soybean AQ. This study also explores the application of soybean AQ in gluten-free crackers, where it was used to adjust texture and moisture content. Moreover, AQ is typically prepared by soaking pulse seed in water to remove unpleasant compounds, soften seed, and reduce cooking time [8]. This step requires large vessels for soaking and generates large quantities of wastewater, especially if soaking water is not efficiently reused. Soaking consumes significant energy, especially at larger scales. To produce more eco-friendly AQ processes, Shim et al. (2021) considered the significance of the presoaking process [18].

The hypothesis of this study is that soybean AQ functional properties (foamability/emulsibility) correlate with soybean seed components and physicochemical properties. Therefore, the primary purpose of the current study was to investigate the physicochemical properties and hydration dynamics of different soybean seeds and evaluate possible correlations with AQ emulsion and foaming properties. In addition, soybean AQ components contribute to its functional properties and potential health benefits. Here, AQ components were characterized using nuclear magnetic resonance (NMR) spectroscopy.

## 2. Materials and Methods

### 2.1. Materials and Chemicals

A Kabuli type of chickpea cultivar [*Cicer arietinum* (L.), *ver.* CDC Leader] was used to produce standard AQ for comparison with three soybean cultivars (Backtae, Seoritae, and Jwinunikong). CDC Leader was generously provided by Dr. Bunyamin Ta’ran from the University of Saskatchewan, Crop Development Centre (CDC, Saskatoon, SK, Canada). All Korean soybean samples were purchased from a local grocer, Nonghyup Hanaro Market (Seoul, Korea). The seed was randomly selected and manually cleaned and freed of dust and other foreign material (Figure 1). All seed was stored at room temperature (20–22 °C) until analysis.

### 2.2. AQ Preparation from Seed

AQ samples were prepared from chickpea and three soybean cultivars as described by Shim et al. (2021) [18] and He et al. (2021) [19]. A total of 12 samples (four types of seed soaking water, P0–P3; six AQ samples, 0–3, S0–S3, Table 1) were produced in this study to evaluate the effects of the AQ preparation steps. Water obtained from presoaking seed was used for NMR spectroscopy.

#### 2.2.1. AQ Produced without Presoaking

First, 200-g samples of each type of seed were rinsed with distilled water then mixed with 300 mL of distilled water (1:1.5, *w*/*w*) in 500-mL sealed glass canning jars and cooked in a pressure cooker (Instant Pot^®^ 7 in 1 multi-use programmable pressure cooker, IP-DUO60 V2, 6 quart/liters) at 115–118 °C (an autogenic pressure range of 70–80 kPa) for 90 min. Subsequently, jars of cooked soybean and chickpea were cooled by holding at room temperature for 24 h. Cooled AQ was drained from cooked seed using a stainless-steel mesh kitchen strainer, marked as sample 0, 1, 2, or 3, and stored in a freezer (−20 °C). Each procedure was replicated three times. Subsamples of AQ were frozen to −20 °C then dried in a freeze dryer (FreeZone 12 Liter Console Freeze Dryer with Stoppering Tray Dryer, Labconco Corporation, Kansas City, MS, USA) until the sample temperature rose to −5 °C, indicating the sample had been thoroughly dried.

#### 2.2.2. AQ Produced with Presoaking

First, 200 g of each type of seed were washed with distilled water and presoaked in distilled water at a ratio of 1:3 (*w*/*w*, chickpea) and 1:4 (*w*/*w*, soybean) for 16 h at 4 °C. Each sample of water from presoaking (samples P0–P3) was drained and freeze-dried for NMR analysis. Subsequently, 400 g samples of presoaked seed were rinsed with distilled water and mixed with 400 mL of distilled water in sealed glass canning jars. Seeds were cooked in a pressure cooker as described in Section 2.2.1 except for chickpea AQ, which was cooked for 30 min, as this was previously reported to be the optimum condition for producing chickpea AQ [19]. The four AQ samples produced were labelled S0, S1, S2, or S3.

### 2.3. Seed Physical Properties

The hundred seed weight (*HSW*, g) was determined by randomly selecting and weighing 100 grains from each soybean sample. Seed coat incidence (*SCI*, %) on the entire seed weight was determined by the method of Avola and Patanè (2010) [20] with minor modifications. The seed coat (Figure 1) of 10 seeds was removed after presoaking seeds in distilled water at room temperature (20–22 °C) for 12 h. Then, the seed coat and cotyledons (Figure 1) were dried separately at 65 °C for 4 h, then weighed each hour until a constant weight was recorded. Seed dimension was determined by randomly selecting 10 seeds and then using a micrometer to record seed dimensions in three perpendicular directions. Equation (1) was used to calculate the geometric average of the diameter of an equivalent sphere:(1)$$ED={\left(L\times W\times T\right)}^{1/3}$$
where *ED* is the equivalent dimension (mm); and *L*, *W*, and *T* are the major, minor, and intermediate axes (mm), respectively [21].

The surface area per unit mass of seed (or, specific surface area, *SSA*, mm^2^/mg) and seed coat weight per surface area (namely seed coat thickness, WSA, mg/cm^2^) of a single seed were calculated based on the *ED* and *HSW* value by Equations (2) and (3), respectively:(2)$$SSA=\frac{\pi \times ED^{2}}{HSW\times 10}$$
(3)$$WSA=\frac{HSW\times SCI\times 1000}{\pi \times ED^{2}}$$

### 2.4. Seed Hydration Kinetics

Hydration kinetic tests were performed by the method of Avola and Patanè (2010) [20] with minor modification. Soybean seed was soaked at room temperature (20–22 °C) and weighed periodically to determine the kinetics of water uptake. Ten seeds were transferred to a 200 mL beaker, which contained 150 mL of deionized water or aqueous solutions of 0.5% (*w*/*v*) NaCl or NaHCO_3_. Beakers were held at a constant temperature of 22 °C. Each hour up to the seventh hour, then at 24 h after initial imbibition, the seed was drained and then weighed after free water was absorbed with a delicate task tissue wipe (Kimberly-Clark Inc., Mississauga, ON, Canada). A clean tissue wipe was used for each weighing to avoid contamination with solutes or water. A two-parameter asymptotic Equation (4) was used to model water uptake kinetics (SigmaPlot 12.0; Systat Software, Inc., San Jose, CA, USA):(4)$$H_{t}=H_{max}\times \left(1-e^{-kx}\right)$$
where *H_t_* is the hydration weight (g/seed) after soaking for time t (h), *H_max_* is the asymptote of the curve (to estimate seed weight at full hydration), and *k* is a curve parameter that is related to the initial hydration rate (estimating *H_rate_*).

### 2.5. Color

The color of three cultivars of soybean seed and freshly prepared liquid AQ samples were determined using a Hunter lab ColorFlex spectrophotometer (Hunter Associates Laboratory, Inc., Reston, VA, USA), set on the CIELAB coordinates [*L** (lightness/darkness), *a** (redness/greenness), and *b** (yellowness/blueness)]. The instrument was standardized with black and white tiles (X = 79.1, Y = 83.8, Z = 89.4), and a green tile (*L* = 52.9, *a* = –5.8, *b* = 12.9) as a color check.

### 2.6. NMR Spectroscopy

For aqueous AQ samples, double pulse field gradient spin echo proton (DPFGSE-^1^H-NMR) was conducted to detect and quantify compounds with resonances that were well away from the suppression frequency according to a modified method based on Shim et al. (2018) [12]. To observe anomeric protons of AQ carbohydrate components, ^1^H-NMR spectra were collected from freeze-dried AQ samples that were reconstituted in deuterated water. Proton NMR spectra for both DPFGSE water suppression and samples rehydrated with deuterated water were recorded on a Bruker Avance 500 MHz NMR spectrometer (Bruker BioSpin, Rheinstetten, Germany) with 16 scans per spectrum using a DPFGSE-NMR sequence or standard ^1^D sequence. NMR data collection and analysis were conducted with TopSpin^TM^ 3.2 software (Bruker BioSpin GmbH., Billerica, MA, USA). AQ samples (0.4 mL) were transferred to 5-mm NMR tubes where the samples were mixed with 50 mg of added deuterium oxide (Cambridge Isotope Laboratories Inc., Andover, MA, USA) to provide a solvent lock signal prior to NMR analysis. For ^1^H-NMR, 3-(trimethylsilyl)propionic-2,2,3,3-d4 acid sodium salt (TMSP, EMD Chemicals Inc., Gibbstown, NJ, USA) was used as an internal standard.

### 2.7. Statistical Analysis

All analyses were performed in triplicate to obtain the average and SD values. Data are presented as mean ± SD (*n* = 3). Analytical results were processed with Microsoft Excel 2018. Statistics were implemented through the Statistical Package for the Social Science (SPSS) version 25.0 (IBM Corp., Armonk, NY, USA). Analysis of variance (ANOVA) and Tukey’s tests were used to evaluate the statistical significance of differences in properties and composition. Statistical significance was accepted at *p* < 0.05. The mathematical model parameters used in seed hydration kinetics measurements were estimated using a nonlinear regression procedure performed using SigmaPlot 12 software (Systat Software Inc. San Jose, CA, USA). Model suitability was evaluated using the coefficient of determination (*R*^2^), which indicates the model predictive quality (the higher the value for *R*^2^, the better the goodness of fit, and up to a value of 1 meaning exact fit). The hydration kinetics parameters given by the nonlinear regressions were used to compare soaking treatments, including soaking time, seed type, and soaking solution as variables. The Pearson correlation coefficients (*r*) for the relationships between all characteristics were calculated. The correlations between soybean variety properties (dry seed composition, AQ moisture, AQ yield, AQ emulsion and foaming properties) determined in the previous study [18] and the physical/technical characteristics of these soybeans were investigated.

## 3. Results and Discussion

### 3.1. Physical Characteristics of Dried Seed

The physical properties of seeds of different soybean varieties are shown in Table 2. Significant differences were observed among all physical properties (HSW, ED, SCI, SSA, and WSA) of each cultivar (*p* < 0.05). Seoritae showed heavier (HSW = 32.52 ± 0.93 g) and larger seeds (ED = 8.20 ± 0.72 mm) compared to other soybeans (*p* < 0.05). Jwinunikong, as the smallest seed (HSW = 11.44 ± 0.06 g, ED = 5.68 ± 0.35 mm), had the highest SCI (8.21 ± 0.67%). A higher SCI value reflects a higher ratio of fiber content, which is associated with stronger diffusion resistance and inhibited leaching of soluble solids during soaking and cooking.

The differences in SSA and WSA are reflected in the seed coat behavior during soaking and cooking and influence the rate at which substances leach from seed, which could affect AQ functionality. For instance, Backtae has considerably lower SSA (0.597 ± 0.046 mm^2^/mg) and WSA (7.39 ± 0.38 mg/cm^2^), possibly explaining why AQ made with this variety had the highest dry matter (15.19 g/100 g seeds) and emulsion capacity/stability [18]. Differences in the culinary potential of various chickpea genotypes have been previously reported to be due to differences in seed characteristics [7,20].

### 3.2. Seed Hydration Kinetics

The ability to absorb water during soaking is usually related to seed physical properties. Hydration of the seed coat and swelling of the cotyledon leads to cell wall softening, changes in tissue permeability, shortening of cooking times, and increased mass transfer from the seed to the cooking water. The relationship between the soaking time and the cumulative value of water absorption (Figure 2) was described by a nonlinear iterative regression method with an exponential relationship (Equation (4)). The applied model fitted the experimental data with an *R*^2^ that was greater than 0.94 for the soybean cultivars Backtae and Seoritae. Therefore, a single curve for their water uptake was used with all data combined for these two cultivars. Soaking processes achieved an initial rapid water absorption rate (*H_rate_* = 0.3655 H_2_O g/min), which might reflect rapid water uptake through the hilar fissure [22]. Absorbed water reached 90% of the total water absorption weight after soaking for 6.3 h. Subsequently, the water absorption rate declined until the hydrated seed weight was 2.4-fold greater than before hydration, where total hydration reached saturation at 1.40 g H_2_O g/dw (*H_max_*). This value is similar to the maximum water absorption of soybean (*G. max*) (130%) reported by Li et al. (2019) [23]. Soybean hydration exhibited the same behavior reported by other authors [22,24,25,26]. The water content of chickpea seeds used as a control exceeded 90% of the total water absorption after 6 h [10].

On the contrary, Jwinunikong absorbed water more slowly (0.1822 g H_2_O g/min), and after soaking over double the amount of time (12.6 h) when compared with Backtae and Seoritae, the absorbed water reached 90% of the total weight after water absorption. Here, the hydration equilibrium reached 1.491 g H_2_O/g seed dw (*H_max_*). These variables may be attributed to the different water diffusivity in soybean and the cultivar physicochemical characteristics (chemical composition, seed size and density, SCI, SSA, etc.) [22].

The statistical analysis in Table 2 indicated no significant differences between the *H_max_* values of the three soybeans in all the soaking solutions, but the hydration rate (*H_rate_*) showed a significant difference (*p <* 0.05). Interestingly, the *H_max_* of soybean seed soaked in NaCl solution was slightly lower than that of H_2_O (Table 3). In contrast, the immersion of seeds in NaHCO_3_ solution relatively increased *H_max_* in both Backtae and Seoritae but decreased *H_max_* in Jwinunikong. The increased *H_max_* observed in NaHCO_3_ solution may be due to the interaction of carbonate ions with the biopolymer in the cotyledons cells, resulting in molecular unwinding [27]. Moreover, soaking seed in NaCl and NaHCO_3_ solution decreased *H_rate_* for Backtae but increased *H_rate_* for both Seoritae and Jwinunikong. In Li’s study [23], faster water absorption was also observed when soaking soybean in 0.5% NaHCO_3_ solution. These results indicate that the process of water absorption varied with soaking solution conditions. In contrast, the water absorption equilibrium values (*H*_max_) remained independent and were not affected by these factors.

### 3.3. Color Analysis

The color of AQ differed depending on the seed type and pre-soaking vs. not-soaked treatment (Figure 3, Table 4). P0–P3, which were seed soaking water, were translucent liquid samples leached from the seed. AQ produced from CDC Leader (0, S0) and Backtae (1, S1) were beige to dark yellow solutions derived from the Kabuli chickpea cultivar (*C. arietinum* L.) and yellow soybean [*G. max* (L.) Merr.], respectively. CDC Leader is a Kabuli class chickpea cultivar that has a cream-yellow color mainly arising from carotenoid pigments [28]. Backtae, as a yellow soybean cultivar, has a yellow seed coat and also contains carotenoid pigments with small amounts of chlorophyll and water-soluble vitamins, such as thiamine, niacin, biotin, and pantothenic acid [29]. Xanthophylls, for example, are a group of yellow pigments that were identified in yellow soybean. Samples 0 and 1 had a higher *a** value than that of S0 and S1 thus, they appear redder than the latter. Remarkably, the *b** values of samples 0 and 1 were significantly higher (*p <* 0.05) than those of S0 and S1 (21.47 ± 0.06 vs. 15.55 ± 0.40; 14.43 ± 0.14 vs. 11.33 ± 0.06, respectively), indicating that the color of AQ from CDC Leader and Backtae without presoaking tended toward yellow. This might be due to the higher dry matter content in samples 0 and 1. On the other hand, presoaking could remove some water-soluble pigments and vitamins, so the concentration of pigment and water-soluble vitamins in samples without presoaking were higher [30]. Interestingly, sample 0 was nearly translucent, but sample S0 was a cloudy liquid with higher turbidity. The variation in turbidity is a direct indicator of nutrients (proteins, sugars, pigments, etc.) being released from seed to form AQ during cooking [31]. Therefore, it was assumed that even with prolonged cooking times (sample 0, 90 min vs. sample S0, 30 min), AQ produced from chickpea without presoaking was not as fully cooked as presoaked samples.

On the other hand, AQ produced from Seoritae in samples 2 and S2 and Jwinunikong in samples 3 and S3 was cloudy and dark brown to black from large and small black soybeans, respectively. Seoritae has a black seed coat, and its cotyledon is green (Figure 1). Jwinunikong also has a black seed coat, and its cotyledon is greenish (Figure 1), and the plants are characterized by a higher abundance of isoflavones compared with other soybean cultivars [32]. Therefore, it was not surprising that AQ produced from black seed-coated soybeans had a dark brown color and high turbidity. This dark brown color might be caused by anthocyanins leaching from the black large and small soybeans seed coats during cooking [33]. Anthocyanins are water-soluble flavonoid pigments as the main subclass of polyphenol compounds [34]. Moreover, the *L*^*^ values of samples S2 and S3, which represent the brightness of AQ, were significantly higher (*p <* 0.05) than that of samples 2 and 3, indicating that presoaking increased AQ brightness. Presoaking can extract dark pigment components of Seoritae and Jwinunikong that are partly or entirely solubilized in the soaking solution, leading to an increase in AQ brightness.

### 3.4. Correlation Analysis

The correlations among the characteristics of different soybean varieties (dried seed composition, aqueous AQ moisture, emulsion, and foaming properties, freeze-dried AQ yield) determined in the previous study [18], and the physical/technological properties of these soybeans in the current study are shown in Table 5 (Pearson correlation coefficients, *r*, followed with * and **, indicate significance for *p* < 0.05 and 0.01, respectively). Some significant correlations were found for aqueous AQ produced without presoaking (1–3). For example, the moisture content and emulsion stability (ES) showed a negative correlation (*r* = −0.998, *p <* 0.05). This relationship was also observed in chickpea AQ with the presoaking process by He et al. (2019) [10], suggesting that higher dry matter content in AQ results in better emulsion stability. In addition, AQ emulsion capacity (EC) showed a negative correlation (*r* = −0.997, *p <* 0.05) with soybean SCI and a close positive correlation (*r* = 1.000, *p <* 0.01) with soybean fat content. The foaming capacity (FC) of AQ (1–3) showed a positive correlation with soybean ash content (*r* = 0.999, *p <* 0.01). The foam stability (FS) of AQ (1–3) showed a negative (*r* = −1.000, *p <* 0.01) and positive correlation (*r* = 0.997, *p <* 0.05) with soybean SCI and fat content, respectively. For AQ produced after presoaking (S1–S3), the emulsion capacity was inversely correlated with soybean carbohydrate (*r* = –0.997, *p <* 0.05) and protein contents (*r* = −1.000, *p <* 0.01).

For seeds of the three soybean cultivars, negative correlations were observed between HSW and fiber content (*r* = −0.999, *p <* 0.01) and SCI and fat content (*r* = −0.998, *p <* 0.05), respectively (Table 5). Soybean ED had a very close relationship with *H_rate_* (*r* = 0.999, *p <* 0.01), indicating larger soybeans absorbed water more quickly. Furthermore, SSA was found to have a highly significant negative correlation with WSA (*r* = −1.000, *p <* 0.01). Finally, soybean carbohydrate content correlates with protein content (*r* = 0.998, *p <* 0.05).

AQ functional properties (foaming/emulsification/gelling) are mainly determined by the structure and concentration of AQ composition (protein, water-soluble/insoluble carbohydrates, etc.) [8]. For example, a strong correlation between AQ foaming ability and AQ protein concentration has been found [15]. Therefore, one hypothesis of the current study is that some soybean seeds’ physicochemical and hydration properties might be directly correlated to AQ functional properties. If this assumption is verified, AQ quality can be simply controlled and improved by selecting legume seeds with specific characteristics, such as protein content, carbohydrate content, seed size, and seed weight. On the other hand, another hypothesis is that different AQ functional properties (foaming and emulsion properties in this study) are affected by the same factors. For example, if AQ with higher foaming properties also has higher emulsion properties, AQ optimization can be done to produce one standard soybean AQ product for multiple uses.

However, in this study AQ foaming, and emulsion properties are not always positively correlated with soybean protein and carbohydrate content as expected. This could be due to the different physical characteristics exhibited in the seed of different soybean cultivars and the interaction between these components during cooking and storage [10]. For example, seed coat thickness and seed hardness may influence the dispersal of chemical substances into AQ, even under identical cooking conditions. On the other hand, during the cooking of soybean seed, polysaccharides and proteins can form complexes and coacervates by covalent, Maillard, or electrostatic interaction [8]. These complexes can either build dense viscoelastic interfacial networks at the air/water interface to reduce air bubble coalescence and promote foam stability or stabilize electrostatic repulsion forces between the droplet surfaces and induce stability against flocculation and creaming of emulsions, especially in concert with saponins and phenolic compounds [8,35,36,37]. Meanwhile, no relationship between AQ foaming and emulsion properties can be found. Thus, in the future, two types of commercial AQ products might be expected, with one specifically designed as a foaming agent (in sponge cake, meringue, etc.) and another as an emulsifier (in mayonnaise, salad dressing, etc.). It is also worth noting that the three Korean soybeans in this study are from two different species (Backtae and Seoritae, *G. max* (L.) Merr.; Jwinunikong, *R. nulubilis*), and have two different colors (yellow and black). It is insufficient to explain the relationship between these three cultivars of Korean soybeans, and further studies with these cultivars are required.

The chemical composition of cooking water from soybean varied largely and was determined by soybean cultivar. Composition correlated with seed size, composition, and seed coat and cotyledon structure. In addition, cooking conditions (pressure, temperature, time, etc.) also played an important role in determining composition. Therefore, the relationship between soybean characteristics and AQ functionality is complex. The weak correlation between soybean seeds’ physicochemical properties and AQ functional properties (emulsion/foaming) observed in the current study does indicate that AQ physicochemical properties were not well correlated to AQ functional (emulsion/foaming) properties.

### 3.5. NMR Spectroscopy

NMR spectroscopy was used to detect compounds in aqueous and freeze-dried powders (Figure 4) from seed soaking water, and AQ samples. The resonances of all AQ samples were assigned according to a previous publication [12]. A total of 20 compounds were identified including alcohols (isopropanol, ethanol, methanol), organic acids (lactic acid, acetic acid, succinic acid, citric acid, and malic acid), sugars (glucose, galactose, arabinose, sucrose, raffinose, stachyose), vitamins (choline, phosphocholine), amino acids (alanine, glutamine), and polyphenols (resveratrol, glycitin) in Figure 5, Figure 6 and Figure 7. Most of the DPFGSE-^1^H-NMR spectra of AQ (except AQ sample 2) showed a triplet at 1.2 ppm from the ethanol methyl group (No. 2, Figure 5). The singlets at 1.95 and 3.2 ppm in the DPFGSE-^1^H-NMR spectra of all AQ indicates the presence of acetic acid and choline, respectively (No. 5 and 10, marked black, Figure 5). The small amount of organic acid explains why freshly prepared AQ samples are slightly acidic (pH 6.0–6.3, data not shown). Interestingly, ethanol, acetic acid, and lactic acid detected in AQ were assumed to be produced during steeping of seed prior to canning by Shim et al. (2018) [12], whereas the NMR results in this study revealed that these compounds were also presented in AQ samples produced without presoaking (0–3). It is assumed that these compounds were produced after AQ production and occurred during storage by fermentation [38].

Glycitin (C_22_H_22_O_10_), an isoflavone beta-glucoside with antioxidant effects, is present in the seed coat of black soybean [39], and was found in Jwinunikong AQ samples (P3, S3, 3) and Seoritae AQ sample (2) (No. 14, marked black, Figure 6). According to Shin (2016) [40], the content of isoflavones in cooked small black soybeans significantly increased (7-fold) in uncooked Jwinunikong seed (*p* < 0.001). Different cooking methods affected glycitin contents in Jwinunikong seed: pan-boiled seed (36.43 μg/g), pressure-cooked seed (36.24 μg/g), boiled seed (36.04 μg/g), and steamed seed (27.32 μg/g). The glycitin composition variations depended on the presence and amount of water during cooking, temperature, and cooking time in different cooking conditions. In the current study, the glycitin content in Jwinunikong AQ sample without presoaking (3) was higher than AQ produced after presoaking (S3). This might be a result of the different cooking times between the two preparation methods (with presoaking vs. without presoaking).

A trace amount of resveratrol was shown in P1 and 2 (No. 13, Figure 6). Resveratrol is a polyphenolic compound known to be present in soybean hull with health benefits including antioxidant activity, cardioprotection, anticancer activity, and anti-inflammatory effects [41]. However, this compound has limited solubility in water and is sensitive to heat and oxidation, which might result in isomerization and reduce its functional activity [42]. The tentative attribution of the doublet at 6.4 and 7.4 ppm as resveratrol is based on previous knowledge of the presence of this compound in soy hull. Additional experiments would be required to confirm this assignment. As DPFSE-NMR experiments were performed with just 16 scans, the signal to noise ratio for this peak could be greatly improved by additional scans.

The water-soluble oligosaccharides of all dried AQ powders were mainly composed of sucrose and stachyose (No. 15 and 16, marked in black, Figure 7). Raffinose and arabinose (No. 17 and 18, Figure 7) were only present in chickpea samples (P0, S0, 0). These results confirmed the assumption of Stantiall et al. (2018) [15], who determined the large amount of water-soluble carbohydrates in AQ were low-molecular-weight carbohydrate species and in agreement with previous studies regarding the analysis of sugar losses of cooked seed [43,44,45,46]. Glucose and galactose (No. 19 and 20, Figure 7) were only observed in the presoaking solution (P0–P3), indicating that these monosaccharides leached from the seed during presoaking [44]. Cooking seed in water under high pressure and temperature could destroy these heat-sensitive compounds [44,47,48], thus these compounds were not detected in all AQ samples. The presence of these compounds is thought to be important in the diets of diabetic and galactosemia patients.

The carbohydrate content was reduced in AQ samples produced with presoaking (S1–S3) when compared to those produced without presoaking (1–3) due to the partial loss of low-molecular-weight oligosaccharides/monosaccharides (glucose, galactose, sucrose, raffinose, and stachyose) during soaking. Oligosaccharides cannot be digested by humans due to the absence of an α-galactosidase enzyme. Therefore, oligosaccharides are normally fermented by gut enteric microbiota in the lower intestine, producing gas and gastrointestinal symptoms including flatulence, bloating, diarrhea, and abdominal pain [49,50,51,52,53]. A presoaking step might prove essential for AQ production. On the other hand, stachyose exists in all AQ samples regardless of immersing seed in advance or not in this study. Stachyose and raffinose (especially in chickpea AQ) at lower levels might have health-promoting benefits and, as such, be considered a useful prebiotic by many researchers. Prebiotics stimulate the growth and survival of beneficial intestinal bacteria, such as probiotic *Bifidobacterium* strains, *Lactobacillus bulgaricus*, and *Streptococcus thermophilus* [54,55,56,57].

## 4. Conclusions

In our previous study [18], we reported the emulsification and foaming capacity and stability of AQ extracted from different soybean cultivars as well as soybean seed proximate composition (moisture, ash, crude protein, fat, carbohydrate, and crude fiber). In this study, the 100 seed weight, SCI, seed size, and hydration kinetics of soybean cultivars were measured and compared with previously reported AQ/soybean physicochemical property data [18] to identify possible correlations. The results indicated that soybean physical properties and hydration kinetics were significantly different between the two soybean cultivars *G. max* (L.) Merr. (Backtae and Seoritae) and *R. nulubilis* (Jwinunikong). Weak correlations between soybean seed physicochemical and AQ functional properties were observed related to seed size, composition, seed coat structure, cotyledon structure, and cooking conditions (pressure, temperature, time, etc.). NMR profiles revealed the presence of organic substances in chickpea and soybean AQ and compounds that might have accumulated during soaking due to the action of bacteria (acetic acid and lactic acid). The health-promoting polyphenol glycitin and likely resveratrol, as well as other nutrients (sugars, vitamins, and amino acids) were found in soybean AQ. NMR results also verified a reduction in the concentration of some oligosaccharides/monosaccharides (glucose, galactose, sucrose, raffinose, and stachyose) during presoaking, which, in part, explains differences observed in soybean AQ foaming/emulsification properties where AQ was produced with or without presoaking [18]. This information will allow the development of practices to produce standard commercial soybean AQ products and may help consumers select nutritive products with superior or consistent utility. Further research is needed to determine the chemical composition (protein, fat, soluble fiber, phenolic compounds, etc.) of soybean AQ and evaluate correlations between all chemical characteristics and AQ functional properties. Therefore, soybean AQ provides a new vegan additive option to improve food texture and quality and at the same time adds a unique nutrient substance to the human diet.

## Figures and Tables

**Figure 1 foods-10-02589-f001:**
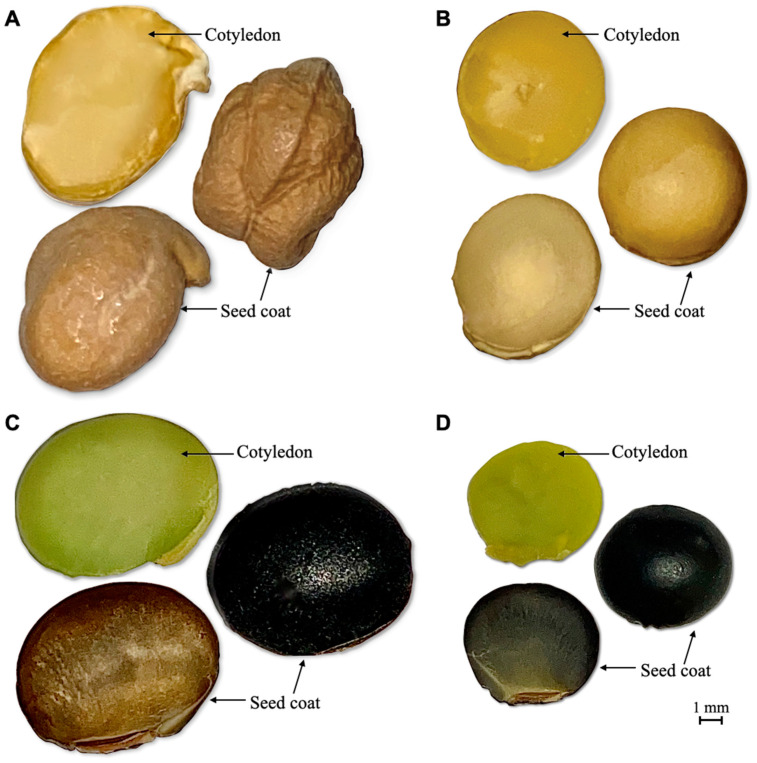
Anatomical structures of (**A**) CDC Leader, (**B**) Backtae, (**C**) Seoritae, and (**D**) Jwinunikong with different seed coat colors. Hand-cut sections of seeds (top and bottom) mounted in distilled water. Top: cotyledons with the seed coat removed; middle: whole seeds; bottom: seed coats (hulls) with the cotyledon removed. Images were obtained (×500 magnification) with a Canon Eos 300D digital camera mounted on a Zeiss Stemi SV 11 light microscope. The images were subsequently processed in Adobe Photoshop 9.0 software. The scale bar is 1 mm.

**Figure 2 foods-10-02589-f002:**
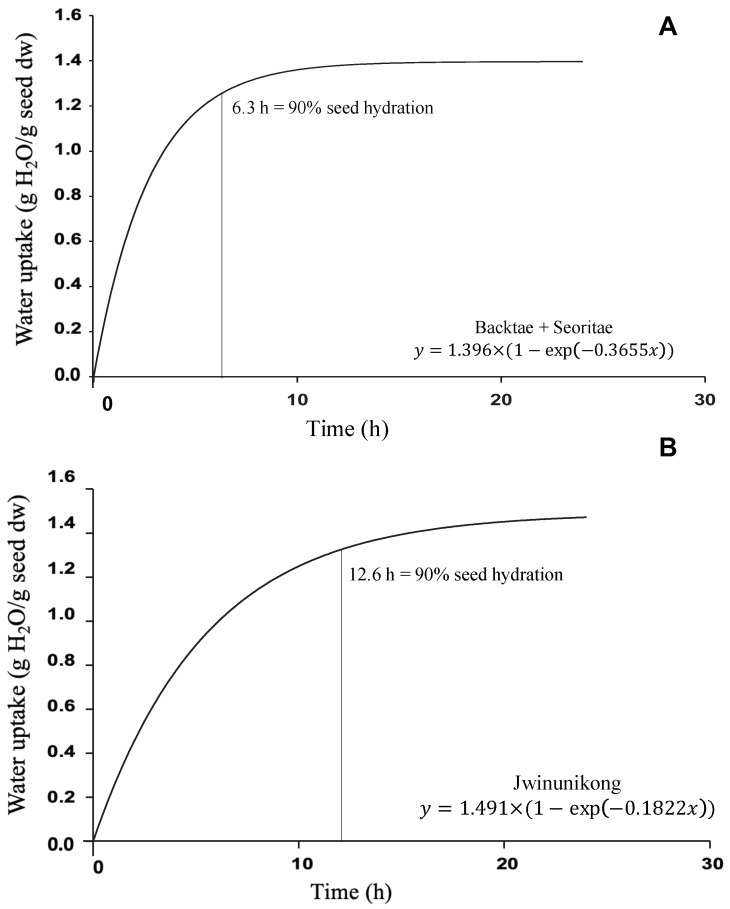
Water absorption kinetics of (**A**) *G. max* (L.) Merr., *ver.* Bactae, Seoritae and (**B**) *R. nulubilis*, *ver.* Jwinunikong. A common curve fitted Backtae and Seoritae.

**Figure 3 foods-10-02589-f003:**
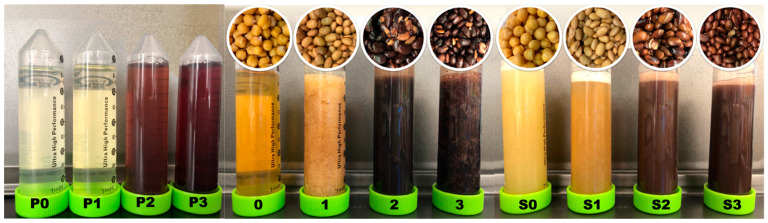
Presoaking water and AQ samples separated from seeds. Top (L–R, circled pictures): the remaining seeds after separation of the AQ liquids after cooking, w/o presoaking AQ samples (0–3), and w/presoaking AQ samples (S0–S3); Bottom (L–R, tube pictures): presoaking water (P0–P3); AQ liquids, w/o presoaking AQ samples (0–3), and w/presoaking AQ samples (S0–S3). AQ samples modified from Shim et al. (2021) [18].

**Figure 4 foods-10-02589-f004:**
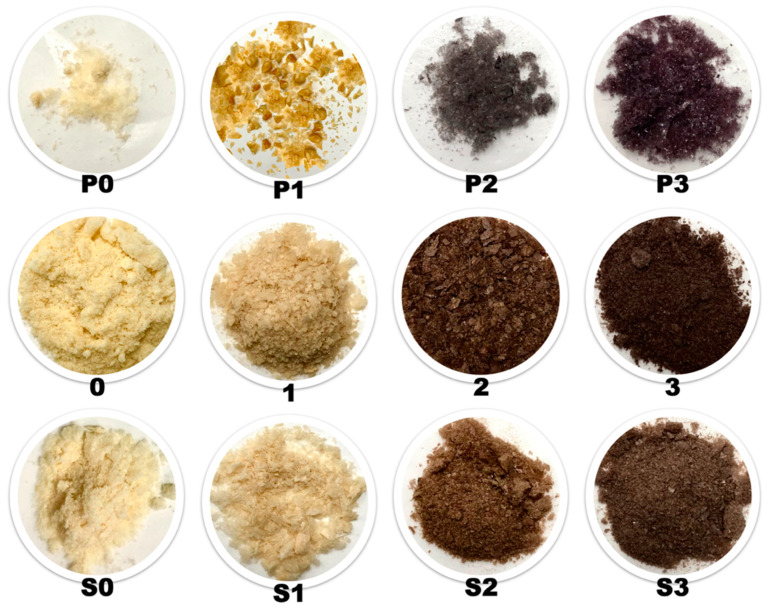
AQ powder after lyophilization: Top (left to right): presoaking water samples (P0–P3); middle: w/o presoaking AQ samples (0–3); and bottom: w/presoaking AQ samples (S0–S3).

**Figure 5 foods-10-02589-f005:**
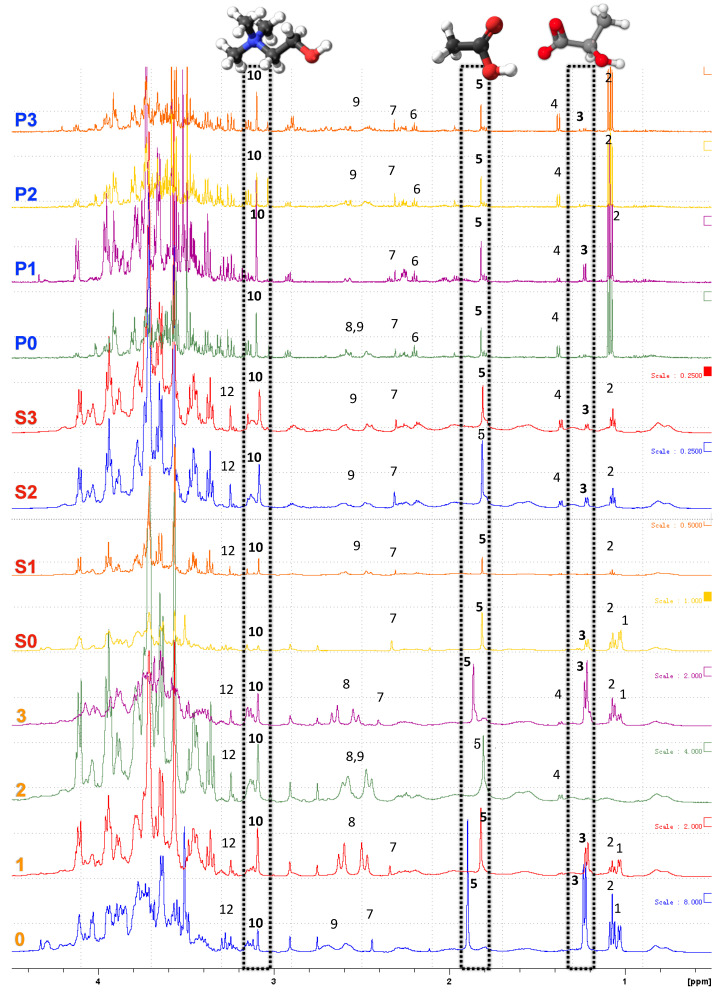
DPFGSE-^1^H-NMR spectra of seed soaking waters/aqueous AQ samples after different treatments. Assigned peaks arise from the presence of 1, isopropanol; 2, ethanol; 3, lactic acid; 4, alanine; 5, acetic acid; 6, glutamine; 7, succinic acid; 8, citric acid; 9, malic acid; 10, choline; 11, phosphocholine; 12, methanol (Cont’d).

**Figure 6 foods-10-02589-f006:**
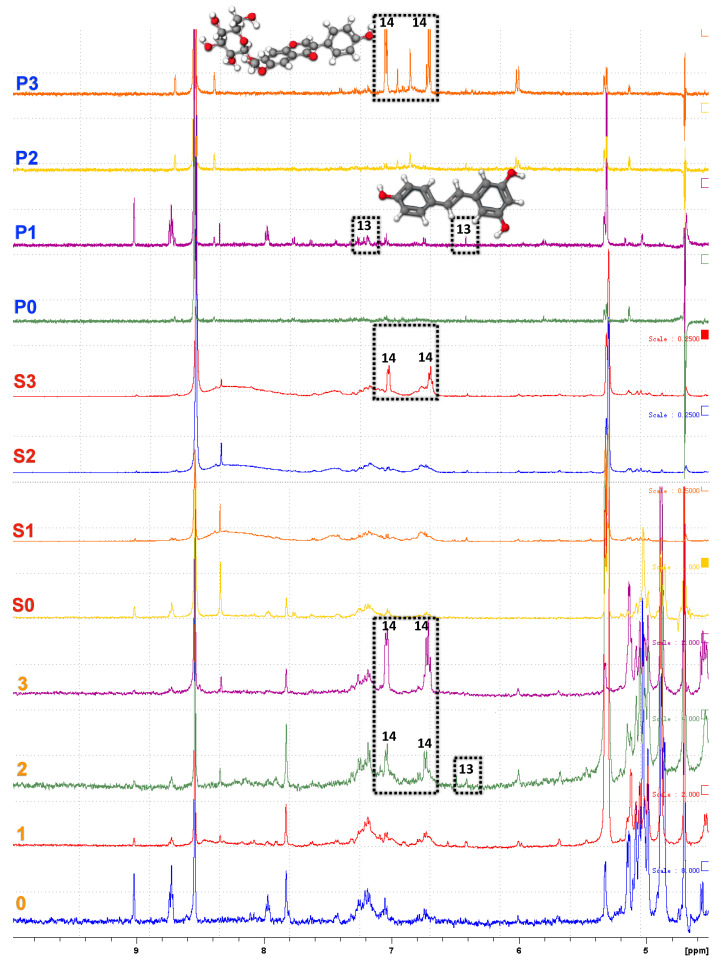
DPFGSE-^1^H-NMR spectra of seed soaking waters/aqueous AQ samples after different treatments. Assigned peaks arise from the presence of 13, resveratrol; 14, glycitin.

**Figure 7 foods-10-02589-f007:**
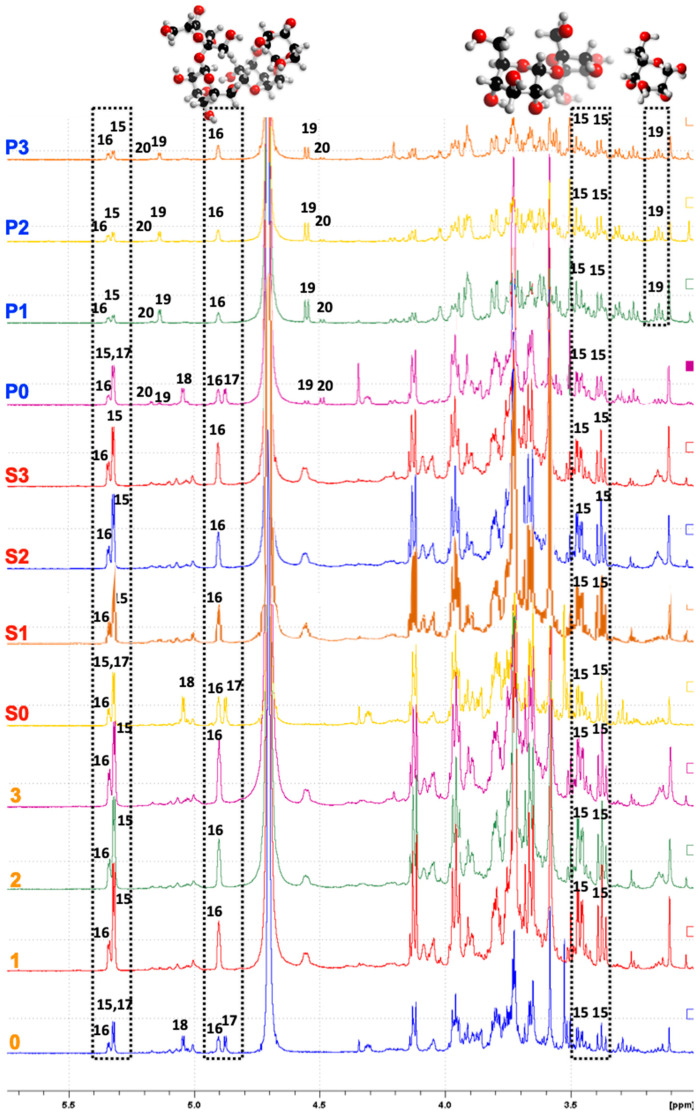
^1^H-NMR spectra of seed soaking waters/AQ powder samples after different treatments. Assigned peaks arise from the presence of 15, sucrose; 16, stachyose; 17, raffinose; 18, arabinose; 19, glucose; 20, galactose.

**Table 1 foods-10-02589-t001:** AQ preparation conditions.

	Species/Cultivar (Code)	Canadian Chickpea ^1^	Korean Soybeans ^2^		
Group		*Cicer arietinum* (L.)	*Glycine max* (L.) Merr.		*Rhynchosia nulubilis*
		CDC Leader	Backtae	Seoritae	Jwinunikong
Presoaking water		P0	P1	P2	P3
w/o Presoaking AQ ^3^		0	1	2	3
w/Presoaking AQ ^4^		S0	S1	S2	S3

^1^ Canadian chickpea as a control (light gray column): CDC Leader; ^2^ Korean soybeans: (1) Backtae: yellow soybean, (2) Seoritae: large black soybean, (3) Jwinunikong: small black soybean; ^3^ w/o: without; ^4^ w/: with.

**Table 2 foods-10-02589-t002:** Characteristics of the dried seeds.

Characteristics	Unit	CDC Leader ^1^	Backtae	Seoritae	Jwinunikong
*Physical*					
HSW	g	42.90 ± 0.31 ^a^	27.97 ± 0.46 ^c^	32.52 ± 0.93 ^b^	11.44 ± 0.06 ^d^
ED	mm	8.49 ± 0.21 ^a^	7.29 ± 0.28 ^b^	8.20 ± 0.72 ^a^	5.68 ± 0.35 ^c^
SCI	%	3.89 ± 0.30 ^c^	5.65 ± 0.19 ^b^	7.79 ± 1.29 ^a^	8.21 ± 0.67 ^a^
SSA	mm^2^/mg	0.528 ± 0.03 ^b^	0.597 ± 0.046 ^b^	0.654 ± 0.111 ^b^	0.890 ± 0.106 ^a^
WSA	mg/cm^2^	7.39 ± 0.38 ^c^	9.51 ± 0.74 ^b^	12.23 ± 2.29 ^a^	9.35 ± 1.22 ^b^
*Technological*					
*H_max_*	g (H_2_O g/dw)	1.036 ± 0.022 ^b^	1.436 ± 0.017 ^a^	1.344 ± 0.040 ^a^	1.491 ± 0.144 ^a^
*H_rate_*	g (H_2_O g/min)	0.3537 ± 0.0064 ^b^	0.3289 ± 0.0282 ^b^	0.4263 ± 0.0481 ^a^	0.1822 ± 0.0282 ^c^
*R^2^*	NA	0.9948	0.9413	0.9478	0.9528

Values (mean ± SD) within rows followed by the same letter do not statistically differ at *p* < 0.05 by Tukey’s test. ^1^ CDC Leader’s data were modified from He et al. (2019) [10] as a control (light gray column); HSW, Hundred seed weight; ED, Equivalent dimension; SCI, Seed coat incidence; SSA, Specific surface area; WSA, Seed coat weight per surface area; *H_max_*, Hydration capacity (t = ∞); *H_rate_*, Hydration rate; NA, not available.

**Table 3 foods-10-02589-t003:** Kinetic constants of the nonlinear regression analysis for soybean seeds hydration.

Hydration Solution	*H_max_* g H_2_O/g Seed dw	*H_rate_* g/min	*R^2^*
*Backtae*			
H_2_O	1.429	0.3626	0.9524
NaCl	1.422	0.3199	0.9314
NaHCO_3_	1.455	0.3092	0.9486
*Seoritae*			
H_2_O	1.327	0.3911	0.9350
NaCl	1.318	0.4824	0.9476
NaHCO_3_	1.392	0.4106	0.9741
Backtae and Seoritae	1.396	0.3655	0.9410
*Jwinunikong*			
H_2_O	1.662	0.1527	0.9709
NaCl	1.403	0.2070	0.9584
NaHCO_3_	1.422	0.1930	0.9483
All data combined	1.432	0.2798	0.8850

*H_max_*, Max hydration capacity; *H_rate_*, initial hydration rate.

**Table 4 foods-10-02589-t004:** Color parameters ^1^ of all samples used for AQ production.

Group	*L**	*a**	*b**
*Seed*			
CDC Leader	62.98 ± 0.19 ^b^	8.54 ± 0.02 ^a^	16.77 ± 0.10 ^c^
Backtae	66.30 ± 0.19 ^a^	6.77 ± 0.10 ^b^	24.17 ± 0.12 ^a^
Seoritae	34.53 ± 0.11 ^e^	−0.64 ± 0.08 ^h^	−4.42 ± 0.06 ^kl^
Jwinunikong	34.76 ± 0.10 ^e^	–0.61 ± 0.06 ^h^	−4.89 ± 0.07 ^l^
*Presoaking water*			
P0	21.13 ± 0.13 ^i^	−0.35 ± 0.11 ^h^	−3.56 ± 0.22 ^ij^
P1	19.13 ± 0.37 ^k^	−0.70 ± 0.27 ^h^	−3.91 ± 0.41 ^jk^
P2	21.20 ± 0.44 ^i^	0.04 ± 0.06 ^gh^	−3.29 ± 0.16 ^i^
P3	19.97 ± 0.28 ^j^	0.16 ± 0.08 ^fg^	−3.50 ± 0.08 ^ij^
*w/o Presoaking AQ*			
0	46.35 ± 0.06 ^c^	2.55 ± 0.01 ^d^	21.47 ± 0.06 ^b^
1	33.08 ± 0.22 ^f^	5.27 ± 0.17 ^c^	14.43 ± 0.14 ^e^
2	16.10 ± 0.04 ^m^	2.05 ± 0.24 ^de^	−1.75 ± 0.10 ^h^
3	17.08 ± 0.15 ^l^	2.20 ± 0.03 ^de^	−1.26 ± 0.12 ^h^
*w/Presoaking AQ*			
S0	39.30 ± 0.14 ^d^	0.79 ± 0.02 ^fg^	15.55 ± 0.40 ^d^
S1	38.99 ± 0.24 ^d^	1.25 ± 0.17 ^ef^	11.33 ± 0.06 ^f^
S2	27.61 ± 0.40 ^g^	4.76 ± 0.12 ^c^	1.00 ± 0.18 ^g^
S3	26.19 ± 0.54 ^h^	5.12 ± 0.25 ^c^	1.08 ± 0.07 ^g^

^a–l^ Values followed by different letters within a column are significantly different (*p <* 0.05) according to Tukey’s test. *L*, brightness/darkness; *a*, (+) redness/(−) greenness; and *b*, (+) yellowness/(−) blueness. ^1^ Color parameters’ data modified from Shim et al. (2021) [18].

**Table 5 foods-10-02589-t005:** Correlation coefficients among the physical, chemical, and hydration attributes for the three soybean cultivars and AQ yield, emulsion and foaming capacity, emulsion, and foaming stability.

	AQ MO ^1^	ES ^1^	EC ^1^	FC ^1^	FS ^1^	HSW	SCI	ED	SSA	WSA	CAR ^1^	PRO ^1^	Fiber ^1^	Fat ^1^	Ash ^1^	*H* * _max_ *	*H* * _rate_ *
AQ YI (S1–S3) **^1^**	0.983	0.911	0.953	0.209	−0.472	0.536	−0.996	0.396	−0.815	−0.223	−0.926	−0.948	−0.578	0.988	0.645	−0.104	0.357
AQ YI (1–3) **^1^**	−0.952	−0.698	0.201	0.748	0.108	−0.828	−0.122	−0.907	0.555	−0.981	−0.404	−0.345	0.799	0.184	0.782	0.991	−0.924
AQ MO (S1–S3) **^1^**		0.971	0.881	0.385	−0.302	0.682	−0.961	0.558	−0.908	−0.040	−0.841	−0.873	−0.718	0.942	0.493	−0.285	0.522
AQ MO (1–3) **^1^**		−0.998 *	−0.513	0.103	−0.592	−0.989	0.581	−0.953	0.971	−0.591	0.320	0.380	0.995	−0.528	0.156	0.820	−0.940
ES (S1–S3) **^1^**			0.743	0.594	−0.066	0.837	−0.868	0.740	−0.982	0.199	−0.687	−0.732	−0.863	0.835	0.271	−0.506	0.711
ES (1–3) **^1^**			0.560	−0.048	0.636	0.979	−0.625	0.935	−0.983	0.545	−0.372	0.062	−0.988	0.575	−0.101	−0.787	0.920
EC (S1–S3) **^1^**				−0.097	−0.717	0.255	−0.977	0.099	−0.602	−0.508	−0.997 *	−1.000 **	−0.303	0.989	0.846	0.202	0.057
EC (1–3) **^1^**				0.801	0.996	0.382	−0.997 *	0.230	−0.703	−0.389	−0.977	−0.989	−0.427	1.000 **	0.768	0.070	0.189
FC (S1–S3) **^1^**					0.764	0.937	−0.117	0.981	−0.736	0.907	0.176	0.113	−0.919	0.054	−0.613	−0.994	0.988
FC (1–3) **^1^**					0.741	−0.248	−0.750	−0.398	−0.136	−0.863	−0.909	−0.881	0.199	0.790	0.999 **	0.654	−0.437
FS (S1–S3) **^1^**						0.491	0.552	0.623	−0.126	0.965	0.770	0.728	−0.447	−0.604	−0.978	−0.828	0.655
FS (1–3) **^1^**						0.467	−1.000 **	0.321	−0.767	−0.300	−0.953	−0.970	−0.511	0.997 *	0.704	−0.024	0.281
HSW							−0.455	0.987	−0.926	0.703	−0.177	−0.240	−0.999 **	0.398	−0.299	−0.895	0.980
SCI								−0.308	0.758	0.314	0.957	0.974	0.499	−0.998 *	−0.714	0.010	−0.072
ED									−0.854	0.807	−0.020	0.083	−0.978	0.248	−0.447	−0.955	0.999 **
SSA										−0.382	0.536	0.589	0.944	−0.715	−0.084	0.661	−0.832
WSA											0.575	0.522	−0.667	−0.373	−0.889	−0.946	0.831
CAR **^1^**												0.998 *	0.226	−0.973	−0.886	−0.279	0.071
PRO **^1^**													−0.288	−0.986	−0.854	−0.217	−0.041
Fiber **^1^**														−0.443	0.251	0.872	−0.969
Fat **^1^**															0.756	0.052	0.207
Ash **^1^**																0.693	−0.484
*H_max_*																	−0.966

Pearson correlation coefficients (*r*) for the relationships between all characteristics were calculated. AQ, aquafaba; YI, yield; MO, moisture; ES, emulsion stability; EC, emulsion capacity; FC, foaming capacity; FS, foaming stability; HSW, 100 seed weight; ED, equivalent dimension; SCI, seed coat incidence; SSA, specific surface area; WSA, weight of seed coat per surface area; CAR, carbohydrate; PRO, protein; *H_max_*, max hydration capacity; *H_rate_*, initial hydration rate. *^,^ ** indicate significant for *p* < 0.05 and 0.01, respectively. ^1^ Data modified from Shim et al. (2021) [18].

## Data Availability

The data of the current study are available from the corresponding author on reasonable request.

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
