# Peer review of "Aquafaba from Korean Soybean II: Physicochemical Properties and Composition Characterized by NMR Analysis"

_foods, 2021, doi:10.3390/foods10112589_

Round 1

Reviewer 1 Report

This work describes the physical properties of seeds of chickpea and three soybean varieties and the hydration kinetics. Also, the correlation analysis between soybean components and aquafaba physical and technological properties are presented.  In addition, the NMR spectroscopy was used to identify 20 compounds of legume soaking water/AQ samples.

The manuscript is concisely and clearly written, the presentation of data is good and well discussed.  This work is interesting, very sound and the rationale is clear and appropriate.

However, the NMR section raises some doubts. For example, in Figure 6, signals for resveratrol and glycitine are identified. For resveratrol, the signals are very small and are almost indistinguishable from the background.

Author Response

Manuscript ID foods-1411509 entitled “Aquafaba Derived from Korean Soybean Cultivars II: Physicochemical Properties and Compositions Characterized by NMR analysis”

Thank you for your patience. We have revised our manuscript (Manuscript ID: foods-1411509) according to the reviewers’ comments/suggestions and would like to thank all reviewers for their critical feedback in making this manuscript more polished. We have listed Reviewer 1’s comments and answered them in sequence. In addition to the changes made to the manuscript as recommended by the reviewers, we have also tried to improve the manuscript’s structure. We appreciate the reviewers’ thoughtful comments and critiques and hope this response addresses the overall quality of this manuscript for publication.

Responses to Reviewer 1 Comments and Suggestions:

This work describes the physical properties of seeds of chickpea and three soybean varieties and the hydration kinetics. Also, the correlation analysis between soybean components and aquafaba physical and technological properties are presented. In addition, the NMR spectroscopy was used to identify 20 compounds of legume soaking water/AQ samples.

The manuscript is concisely and clearly written, the presentation of data is good and well discussed.  This work is interesting, very sound and the rationale is clear and appropriate.

However, the NMR section raises some doubts. For example, in Figure 6, signals for resveratrol and glycitine are identified. For resveratrol, the signals are very small and are almost indistinguishable from the background.

Response: The attribution of the doublet at 6.4 ppm and 7.4 ppm as resveratrol is based on previous knowledge of the presence of this compound in soy hull. The reviewer is correct that the signals are almost indistinguishable from the background. The signal-to-noise ratio of these spectra is determined by the number of scans (16) which was determined before we noted this peak and assigned it as resveratrol. We do not have the option of conducting a greater number of scans to improve the signal at this time. The discussion has been amended accordingly in lines (400-404):

“The attribution of the doublet at 6.4 and 7.4 ppm as resveratrol is based on previous knowledge of the presence of this compound in soy hull. Additional experiments would be required to confirm this assignment. As DPFSE-NMR experiments were performed with just 16 scans the signal to noise ratio for this peak could be greatly improved by additional scans.”

On the other hand, the diagnostic signal for glycitin is the doublet at 6.7 ppm and 7.0 ppm, which can be differentiated from the resveratrol signals.

In addition, we have revised the title of this manuscript to be consistent with the title of the first accepted manuscript.

The previous manuscript (Manuscript ID: foods-1372132, accepted on October 9th, 2021): Aquafaba from Korean Soybean I: A Functional Vegan Food Additive.

This manuscript (Manuscript ID: foods-1411509): Aquafaba from Korean Soybean II: Physicochemical Properties and Composition Characterized by NMR analysis.

Reviewer 2 Report

The article is interesting and shows that the differences in physical properties between different soybean varieties are reflected in seed coat behavior during soaking and cooking and may explain changes in AQ properties and the the dry and emulsion capacity/stability. NMR spectroscopy was used to detect compounds in aqueous and freeze-dried pow- der of legume soaking water/AQ samples. A total of 20 compounds were identified including alcohols, organic acids, sugars, vitamins), amino and polyphenols which shows the importance of achievement of chickpea and soybean AQ.

The study was well conducted, the results are interesting and well-presented but:

  • Very confusing introduction with important concepts, but very mixed. It speaks of beans as well as soy. Must be revised

  • If this is one of the objectives of the study is “In addition, soybean AQ components that contribute to its functional properties and potential health benefits were characterized using nuclear magnetic resonance (NMR) spectroscopy” ..... it cannot be said in the past as if it has already been done.

  • Almost half of the conclusions are a description of what was done and are not conclusions. need review

Author Response

Manuscript ID foods-1411509 entitled “Aquafaba Derived from Korean Soybean Cultivars II: Physicochemical Properties and Compositions Characterized by NMR analysis”

Thank you for your patience. We have revised our manuscript (Manuscript ID: foods-1411509) according to the reviewers’ comments/suggestions and would like to thank all reviewer 2’s for their critical feedback in making this manuscript more polished. We have listed Reviewers’ comments and answered them in sequence. In addition to the changes made to the manuscript as recommended by the reviewers, we have also tried to improve the manuscript’s structure. We appreciate the reviewers’ thoughtful comments and critiques and hope this response addresses the overall quality of this manuscript for publication.

Responses to Reviewer 2 Comments and Suggestions:

The article is interesting and shows that the differences in physical properties between different soybean varieties are reflected in seed coat behavior during soaking and cooking and may explain changes in AQ properties and the the dry and emulsion capacity/stability. NMR spectroscopy was used to detect compounds in aqueous and freeze-dried pow- der of legume soaking water/AQ samples. A total of 20 compounds were identified including alcohols, organic acids, sugars, vitamins), amino and polyphenols which shows the importance of achievement of chickpea and soybean AQ.

The study was well conducted, the results are interesting and well-presented but:

Very confusing introduction with important concepts, but very mixed. It speaks of beans as well as soy. Must be revised

Response: In the introduction section, “beans” were revised to soybeans throughout. This paragraph has been revised to improve the quality of this manuscript.

If this is one of the objectives of the study is “In addition, soybean AQ components that contribute to its functional properties and potential health benefits were characterized using nuclear magnetic resonance (NMR) spectroscopy” ..... it cannot be said in the past as if it has already been done.

Response: This sentence was rewritten as two sentences and the tense was changed to increase clarity in lines (81-83):

“In addition, soybean AQ components contribute to its functional properties and potential health benefits. Here AQ components are characterized using nuclear magnetic resonance (NMR) spectroscopy.

Almost half of the conclusions are a description of what was done and are not conclusions. need review
Response:
The conclusions section was recast to clearly reflect conclusions of the study in lines (463-469, 471-476).

Reviewer 3 Report

I have reviewed the manuscript titled: Aquafaba derived from Korean soybean cultivars II: physicochemical properties and compositions characterized by NMR analysis

This article aims to correlate aquafaba composition with physical properties and seed hydration properties of three Korean soybean cultivars. The information of this work is useful and relevant and there are several physicochemical properties such as hundred seed weight, seed coat incidence, equivalent dimension, specific surface area, seed coat thickness informations of the manuscript that could be adapted by vegetarian-based ingredients with high utility along with meat substitutes in re replacing eggs as emulsifier in the future. The composition, aquafaba yeild, emulsion stability, emulsion capacity, foaming capacity, foaming stability data of soybean could not be found in this manuscript and the previous published paper (ID: Foods (2021)-1372132) was not found on internet. Part of Table 5 results were not mentioned in materials and methods but they were discussed in this paper. The article is innovative, it contains original and interesting information to understand the correlation among aquafaba  emulsion properties and the physical, chemical, hydration properties and aquafaba yields of soybeans if part I of this manuscript could be found! A total of 20 compounds of alcohols, organic acids, sugars, vitamins, amino acids, and polyphenols were identified by NMR and showed the presence of substances in chickpea and soybean aquafaba and those compounds that might have accumulated during soaking due to the action of bacteria. Abstract is well written upon and the four legume seeds and 20 compoounds indetified through NMR were mentioned. Introduction is well addressed including healthier and nutritious foods from plant-based proteins in replacing eggs as emulsifiers. The information of aquafaba from chickpea and soybean and related references were introduced in food nutritious and health promotion effects were also included in the introduction.

Materials and methods were well described except aquafaba, emulsion stability, emulsion capacity, foaming capacity, foaming stability, carbohydration, protein, fiber, fat, and ash content of legume at Table 5 were not mention in this manuscript.

This article would be improved if the authors revise minor revised suggestion at section 3.4 in text section of Table 5 and several compositions and foaming properties in Table 5.

I am not a native English speaker. The manuscript seems have no major mistakes are detected and the manuscript can be easily understood. The results are well discussed except some data in Table 5 seem the reader need to find the previous accepted paper (ID: Foods -1372132) to understand what authors expalin for Table 5.

I enjoyed reading this manuscript; the needs of special groups of emulsifer from legume seeds. This manuscript presents some interesting data and useful NMR analysis information for soybeans and chick peas.

Date of this review

8 October 2021 23:02

Author Response

Manuscript ID foods-1411509 entitled “Aquafaba Derived from Korean Soybean Cultivars II: Physicochemical Properties and Compositions Characterized by NMR analysis”

Thank you for your patience. We have revised our manuscript (Manuscript ID: foods-1411509) according to the reviewers’ comments/suggestions and would like to thank all reviewers for their critical feedback in making this manuscript more polished. We have listed Reviewer 3’s comments and answered them in sequence. In addition to the changes made to the manuscript as recommended by the reviewers, we have also tried to improve the manuscript’s structure. We appreciate the reviewers’ thoughtful comments and critiques and hope this response addresses the overall quality of this manuscript for publication.

Responses to Reviewer 3 Comments and Suggestions:

I have reviewed the manuscript titled: Aquafaba derived from Korean soybean cultivars II: physicochemical properties and compositions characterized by NMR analysis.

This article aims to correlate aquafaba composition with physical properties and seed hydration properties of three Korean soybean cultivars. The information of this work is useful and relevant and there are several physicochemical properties such as hundred seed weight, seed coat incidence, equivalent dimension, specific surface area, seed coat thickness informations of the manuscript that could be adapted by vegetarian-based ingredients with high utility along with meat substitutes in re replacing eggs as emulsifier in the future. The composition, aquafaba yeild, emulsion stability, emulsion capacity, foaming capacity, foaming stability data of soybean could not be found in this manuscript and the previous published paper (ID: Foods (2021)-1372132) was not found on internet.

Response: The previous manuscript (Manuscript ID: foods-1372132) associated with this manuscript was accepted as of October 9th, please check the link below.

https://susy.mdpi.com/user/manuscripts/review_info/27c47e8225b0d242edf2d0ba5097f13b

We have revised the title of this manuscript to be consistent with the title of the first accepted manuscript.

The previous manuscript (Manuscript ID: foods-1372132, accepted): Aquafaba from Korean Soybean I: A Functional Vegan Food Additive.

This manuscript (Manuscript ID: foods-1411509): Aquafaba from Korean Soybean II: Physicochemical Properties and Composition Characterized by NMR analysis.

Part of Table 5 results were not mentioned in materials and methods but they were discussed in this paper. The article is innovative, it contains original and interesting information to understand the correlation among aquafaba emulsion properties and the physical, chemical, hydration properties and aquafaba yields of soybeans if part I of this manuscript could be found! A total of 20 compounds of alcohols, organic acids, sugars, vitamins, amino acids, and polyphenols were identified by NMR and showed the presence of substances in chickpea and soybean aquafaba and those compounds that might have accumulated during soaking due to the action of bacteria. Abstract is well written upon and the four legume seeds and 20 compoounds indetified through NMR were mentioned. Introduction is well addressed including healthier and nutritious foods from plant-based proteins in replacing eggs as emulsifiers. The information of aquafaba from chickpea and soybean and related references were introduced in food nutritious and health promotion effects were also included in the introduction.

Response: Regarding Correlation Analysis, the following sentences were inserted/corrected (lines 318–321, 436). We cited the accepted previous studies [18] in Table 5.

“The correlations among characteristics of different soybean varieties (dried seed composition, AQ moisture, AQ yield, AQ emulsion and foaming properties) determined in the previous study [18], and the physical/technological properties of these soybeans in the current study are shown in Table 5.”

In Table 5, “Pearson correlation coefficients (r) for the relationships between all characteristics were calculated.”

Materials and methods were well described except aquafaba, emulsion stability, emulsion capacity, foaming capacity, foaming stability, carbohydration, protein, fiber, fat, and ash content of legume at Table 5 were not mention in this manuscript.

Response: Regarding Correlation Analysis, the following sentences were inserted in the Materials and Methods (lines 199–202).

“The correlations between soybean variety properties (dry seed composition, AQ moisture, AQ yield, AQ emulsion and foaming properties) determined in the previous study [18] and the physical/technical characteristics of these soybeans were investigated.

This article would be improved if the authors revise minor revised suggestion at section 3.4 in text section of Table 5 and several compositions and foaming properties in Table 5.

Response: Have corrected the same as the above two questions of the reviewer.

I am not a native English speaker. The manuscript seems have no major mistakes are detected and the manuscript can be easily understood. The results are well discussed except some data in Table 5 seem the reader need to find the previous accepted paper (ID: Foods -1372132) to understand what authors expalin for Table 5.

Response: We cited the accepted previous studies [18].

I enjoyed reading this manuscript; the needs of special groups of emulsifer from legume seeds. This manuscript presents some interesting data and useful NMR analysis information for soybeans and chickpeas.

Response: We thank you very much for your high praise.

Reviewer 4 Report

Dear Editors

Respected Authors,

The paper titled “Aquafaba derived form…..” describes the properties of Aquafaba coming from Canadian chickpeas and from three different species of Korean soybeans. The physical properties of seeds and derived water extracts (three samples distinguished according to different production procedure) are evaluated also carrying out the chemical analysis by high resolution NMR. Despite the several measures would lead to some consideration about the four Aquafaba, the research streaming is not so clear.

We list below what should be improved the further production of the authors.

  • According to the title be always aware that physical and chemical properties are somewhat different…as well as processes. Third line in the abstract “…..hydration rates……”
  • The definition of seeds physical parameters is the critical part of the paper; in simple words just three independent variable are present in paragraph 2.3. if I’m not wrong these are size (ED), weight (HSW) and coat plus cotyledon percent in weight (SCI). As the other parameters (SSA and WSA) are dependent from these former ones according to equation 2 and 3 [by the way check it because if I got the math WSA=HSW*SCI/(100*pi*ED^2)]. Provided that the discussion is too complex for an easy reading, the following discussion (par 3.1) should be based just on three over five variables (it can be ok to keep those in table 2). I would suggest to use just W (weight), D (diameter/size on one dimension) and C (coating+ cotyledon %) for the discussion throughout the 3 section, keeping the other entries in the table2.
  • 3.4. third line : the moisture is not a clear concept because samples (0-3) are water based (unless you don’t dry it in some way). Another huge source of confusion is the lack of a short report about the mull parameters mentioned in line 7. These are randomly reported in ref.8 and also in “LWT - Food Science and Technology 132 (2020) 109831” but I could not understand an absolute measurable value except for the emulsion stability which actually looks more similar to a capacity (level gap after whisking the matter…..). This is further complicated by new concepts of foam capacity and stability (are these the same as emulsion…?). Again “negative correlation between SSA and WSA” is mathematically obvious and expected, whereas the conclusions “AQ functional properties (….) are….” Make sense but the whole thing is not even original as this year it was published already (ref 8, 10, DOI: doi:10.3791/56305, with the pending ref.18 which is not available and published yet)
  • Another great dark point is the NMR analysis: why do authors talk about the DPFGSE technique (selected bands) while the spectra look absolutely full-window? Is it used for the water suppression or what? What about the crucial quantification (precision accuracy) of crucial metabolites like sucrose, choline, organic acids and aminoacids? We see authors also used TSP as reference but spectra are not so nice looking so are metabolites really correlated to the other parameters?
  • Page 11 second line: “ …occurring through slow fermentation”
  • Last paragraph page 11: “The overall carbohydrate content was rather reduced in AQ samples with the presoaking process (S1-S3) than in those without the presoaking treatment (1-3)…”. Later, “degusted” is not clear, maybe “digested”.

Author Response

Manuscript ID foods-1411509 entitled “Aquafaba Derived from Korean Soybean Cultivars II: Physicochemical Properties and Compositions Characterized by NMR analysis”

Thank you for your patience. We have revised our manuscript (Manuscript ID: foods-1411509) according to the reviewers’ comments/suggestions and would like to thank all reviewers for their critical feedback in making this manuscript more polished. We have listed Reviewer 4’s comments and answered them in sequence. In addition to the changes made to the manuscript as recommended by the reviewers, we have also tried to improve the manuscript’s structure. We appreciate the reviewers’ thoughtful comments and critiques and hope this response addresses the overall quality of this manuscript for publication.

Responses to Reviewer 4 Comments and Suggestions:

The paper titled “Aquafaba derived form…..” describes the properties of Aquafaba coming from Canadian chickpeas and from three different species of Korean soybeans. The physical properties of seeds and derived water extracts (three samples distinguished according to different production procedure) are evaluated also carrying out the chemical analysis by high resolution NMR. Despite the several measures would lead to some consideration about the four Aquafaba, the research streaming is not so clear.

We list below what should be improved the further production of the authors.

According to the title be always aware that physical and chemical properties are somewhat different…as well as processes. Third line in the abstract “…..hydration rates……”

Response: Have modified the sentence as follows (Line 20)

Physicochemical properties and hydration rates of chickpea (CDC Leader) as a control..

The definition of seeds physical parameters is the critical part of the paper; in simple words just three independent variable are present in paragraph 2.3. if I’m not wrong these are size (ED), weight (HSW) and coat plus cotyledon percent in weight (SCI). As the other parameters (SSA and WSA) are dependent from these former ones according to equation 2 and 3 [by the way check it because if I got the math WSA=HSW*SCI/(100*pi*ED^2)].

Response: We appreciate this thoughtful comment as it helped us identify an error in this manuscript. The equation for SSA (mm2/mg) and WSA (mg/cm2) has been corrected in eqs 2 and 3:

We have confirmed that SSA results in Table 2 are correct and modified the WSA data in Table 2 according to the corrected equation and updated the statistical analysis (Tukey’s test) and correlation analysis (Line 223).

Provided that the discussion is too complex for an easy reading, the following discussion (par 3.1) should be based just on three over five variables (it can be ok to keep those in table 2). I would suggest to use just W (weight), D (diameter/size on one dimension) and C (coating+ cotyledon %) for the discussion throughout the 3 section, keeping the other entries in the table2.

Response: Specific surface area (SSA, surface area per unit mass of seed) and seed coat weight per surface area (WSA) were calculated from seed diameter, seed weight, and seed coat incidence. However, for different soybean cultivars, SSA and WSA were not directly related to any of these three parameters. For example, it is difficult to compare the seed thickness of Backtae seed with larger seed size and weight to smaller Jwinunikong. Therefore, the calculation and discussion of SSA and WSA are necessary. SSA and WSA are very important physical parameters since they influence the seed hydration kinetics, change the substances’ leaching rate/degree from soybean seed into AQ during soaking and cooking, and further affect AQ functionalities (foaming/emulsion properties).

We have revised this section to simplify the discussion. The following sentences have been added in lines 213-221:

“Higher SCI value reflects higher ratio of fiber content which is associated with stronger diffusion resistance and inhibited leaching of soluble solids during soaking and cooking.”

“The differences in SSA and WSA, are reflected in seed coat behavior during soaking and cooking and influence the rate at which substances leach from seed which could affect AQ functionality. For instance, Backtae has considerably lower SSA (0.597± 0.046 mm2/mg) and WSA (7.39 ± 0.38 mg/cm2), possibly explaining why AQ made with this variety had the highest dry matter (15.19 g/100g seeds) and emulsion capacity/stability [18].”

3.4. third line : the moisture is not a clear concept because samples (0-3) are water based (unless you don’t dry it in some way). Another huge source of confusion is the lack of a short report about the mull parameters mentioned in line 7. These are randomly reported in ref.8 and also in “LWT - Food Science and Technology 132 (2020) 109831” but I could not understand an absolute measurable value except for the emulsion stability which actually looks more similar to a capacity (level gap after whisking the matter…..). This is further complicated by new concepts of foam capacity and stability (are these the same as emulsion…?).

Response: In the previous study [18], moisture content, emulsion/foaming properties were measured with freshly prepared aqueous AQ (not dehydrated), while AQ yield was determined based on freeze-dried AQ.

The following sentence has been added at the beginning of 3.4 to clarify the results in this section (lines 325-330):

“The correlations among characteristics of different soybean varieties (dried seed composition, aqueous AQ moisture, emulsion and foaming properties, freeze-dried AQ yield) determined in the previous study [18], and the physical/technological properties of these soybeans in the current study are shown in Table 5 (Pearson correlation coefficients, r, followed with * and **, indicate significance for p < 0.05 and 0.01, respectively). Some significant correlations were found for aqueous AQ produced without presoaking (1–3).”

The significance of the correlation analysis is as follows:

Firstly, the correlation analysis determines if there is any direct relationship between soybean seed characteristics and AQ functional properties. For example, a strong correlation between AQ foaming ability and AQ protein concentration has been found in Ref. 15. Therefore, if we can verify that AQ foaming ability is also correlated to legume seed protein concentration, then AQ quality might be improved by simply selecting legume seed with higher protein content. Secondly, we also determine if different AQ functional properties (foaming and emulsion properties in this study) are affected by the same factors. For example, if AQ with superior foaming properties also have superior emulsion properties, AQ standardization can be done to produce one standard soybean AQ product for multiple uses. If no relationship between these two properties can be found, two products might be expected with one designed to be suitable as a foaming agent (in sponge cake, meringue, etc.) and another as an emulsifier (in mayonnaise, salad dressing, etc.).

Based on the above explanation of the significance of correlation analysis, we have revised this paragraph to clarify our results and discussion. Results and discussion in this section are focused on significant correlation (Table 5, Pearson correlation coefficients, r, followed with * and **, indicate significance for p < 0.05 and 0.01, respectively). and the reasons to explain why the results were not as expected (line 333).

Again “negative correlation between SSA and WSA” is mathematically obvious and expected, whereas the conclusions “AQ functional properties (….) are….” Make sense but the whole thing is not even original as this year it was published already (ref 8, 10, DOI: doi:10.3791/56305, with the pending ref.18 which is not available and published yet)

Response: Original data of aqueous AQ moisture, emulsion capacity/stability, foaming capacity/stability, freeze-dried AQ yield, and soybean proximate composition (carbohydrate, protein, fiber, fat, ash) were published in reference [18]. We have updated this reference information which was fully published on October 13, 2021: Shim, Y.Y.; He, Y.; Kim, J.H.; Cho, J.Y.; Meda, V.; Hong, W.S.; Shin, W.-S.; Kang, S.J.; Reaney, M.J.T. Aquafaba from Korean Soybean I: A Functional Vegan Food Additive. Foods 2021, 10, 2433. https:// doi.org/10.3390/foods10102433

The current study evaluated the relationships among the above characteristics and soybean physical/hydration properties measured. We have revised this paragraph to improve clarify and simplify our results and discussion.

In direct response to the comment by the reviewer that this data is published already, they are correct. However, we have published these two manuscripts as interconnected parts of a greater study. If combined these manuscripts would be a significantly larger work and difficult to publish. We have chosen to divide the work into two manuscripts that share common salient information regarding the starting sample materials.

Another great dark point is the NMR analysis: why do authors talk about the DPFGSE technique (selected bands) while the spectra look absolutely full-window?

Is it used for the water suppression or what? What about the crucial quantification (precision accuracy) of crucial metabolites like sucrose, choline, organic acids and aminoacids?

We see authors also used TSP as reference but spectra are not so nice looking so are metabolites really correlated to the other parameters?

Response: We used the DPFGSE-1H-NMR technique for water suppression (Figures 5, 6) on the original samples to analyze volatile compounds. Figure 7 is the standard 1H-NMR of the freeze-dried samples for carbohydrate profile (Lines 174-185).

We have modified the sentence as follows (Lines 435, 465, 469, 472):

“The water-soluble oligosaccharides of all dried AQ powders were mainly composed of sucrose and stachyose (No. 15 and 16, marked in black, Figure 7).”….

Figure 5. DPFGSE-1H-NMR spectra of seed soaking waters/aqueous AQ samples after different treatments. Assigned peaks arise from the presence of 1, isopropanol; 2, ethanol; 3, lactic acid; 4, alanine; 5, acetic acid; 6, glutamine; 7, succinic acid; 8, citrate; 9, malate; 10, choline; 11, phosphocholine; 12, methanol (Cont’d).

Figure 6. DPFGSE-1H-NMR spectra of seed soaking waters/aqueous AQ samples after different treatments. Assigned peaks arise from the presence of 13, resveratrol; 14, glycitin.

Figure 7. 1H-NMR spectra of seed soaking waters/AQ powder samples after different treatments. Assigned peaks arise from the presence of 15, sucrose; 16, stachyose; 17, raffinose; 18, arabinose; 19, glucose; 20, galactose.

\Page 11 second line: “ …occurring through slow fermentation”

Response: Have modified the sentence as follows (Lines 411, 412)

“It is assumed that these compounds were produced after AQ production and occurred during storage by fermentation.

Last paragraph page 11: “The overall carbohydrate content was rather reduced in AQ samples with the presoaking process (S1-S3) than in those without the presoaking treatment (1-3)…”. Later, “degusted” is not clear, maybe “digested”.

Response: Have modified the sentence as follows (Lines 447-453):

The overall carbohydrate content was reduced in AQ samples produced with presoaking (S1–S3) when compared to those produced without presoaking (1–3) due to the partial loss of low molecular weight oligosaccharides/monosaccharides (glucose, galactose, sucrose, raffinose and stachyose) during soaking. Oligosaccharides cannot be digested by humans due to the absence of the α-galactosidase enzyme and are normally fermented by gut enteric microbiota in the lower intestine, producing gas and gastrointestinal symptoms including flatulence, bloating, diarrhea, and abdominal pain [49–53].

Round 2

Reviewer 4 Report

Dear Editor Respected Authors

I would have some other question of the paper, however I do not like the "back and forth" paper policy, moreover author have replied to many questions and tried to improve the manuscript. From what I can see the other reviewers approve the publication of this manuscript and therefore I join their decision.

Greetings